# MCUNet: Tiny Deep Learning on IoT Devices

Ji Lin[1]    Wei-Ming Chen[1,2]    Yujun Lin[1]    John Cohn[3]    Chuang Gan[3]    Song Han[1]

[1]MIT    [2]National Taiwan University    [3]MIT-IBM Watson AI Lab

https://tinyml.mit.edu

## Abstract

Machine learning on tiny IoT devices based on microcontroller units (MCU) is appealing but challenging: the memory of microcontrollers is 2-3 orders of magnitude smaller even than mobile phones. We propose MCUNet, a framework that jointly designs the efficient neural architecture (TinyNAS) and the lightweight inference engine (TinyEngine), enabling ImageNet-scale inference on microcontrollers. TinyNAS adopts a two-stage neural architecture search approach that first optimizes the search space to fit the resource constraints, then specializes the network architecture in the optimized search space. TinyNAS can automatically handle diverse constraints (*i.e.* device, latency, energy, memory) under low search costs. TinyNAS is co-designed with TinyEngine, a memory-efficient inference library to expand the search space and fit a larger model. TinyEngine adapts the memory scheduling according to the *overall* network topology rather than *layer-wise* optimization, reducing the memory usage by $3.4\times$, and accelerating the inference by $1.7\text{-}3.3\times$ compared to TF-Lite Micro [3] and CMSIS-NN [28]. MCUNet is the first to achieves $>70\%$ ImageNet top1 accuracy on an off-the-shelf commercial microcontroller, using $3.5\times$ less SRAM and $5.7\times$ less Flash compared to quantized MobileNetV2 and ResNet-18. On visual&audio wake words tasks, MCUNet achieves state-of-the-art accuracy and runs $2.4\text{-}3.4\times$ faster than MobileNetV2 and ProxylessNAS-based solutions with $3.7\text{-}4.1\times$ smaller peak SRAM. Our study suggests that the era of always-on tiny machine learning on IoT devices has arrived.

## 1   Introduction

The number of IoT devices based on always-on microcontrollers is increasing rapidly at a historical rate, reaching 250B [2], enabling numerous applications including smart manufacturing, personalized healthcare, precision agriculture, automated retail, *etc*. These low-cost, low-energy microcontrollers give rise to a brand new opportunity of tiny machine learning (TinyML). By running deep learning models on these tiny devices, we can directly perform data analytics near the sensor, thus dramatically expand the scope of AI applications.

However, microcontrollers have a very limited resource budget, especially memory (SRAM) and storage (Flash). The on-chip memory is 3 orders of magnitude smaller than mobile devices, and 5-6 orders of magnitude smaller than cloud GPUs, making deep learning deployment extremely difficult. As shown in Table 1, a state-of-the-art ARM Cortex-M7 MCU only has 320kB SRAM and 1MB Flash storage, which is impossible to run off-the-shelf deep learning models: ResNet-50 [21] exceeds the storage limit by $100\times$, MobileNetV2 [43] exceeds the peak memory limit by $22\times$. Even the int8 quantized version of MobileNetV2 still exceeds the memory limit by $5.3\times$[*], showing a big gap between the desired and available hardware capacity.

Different from the cloud and mobile devices, microcontrollers are bare-metal devices that do not have an operating system. Therefore, we need to jointly design the deep learning model and the inference

---

[*]Not including the runtime buffer overhead (*e.g.*, Im2Col buffer); the actual memory consumption is larger.

**Table 1. Left**: Microcontrollers have 3 orders of magnitude *less* memory and storage compared to mobile phones, and 5-6 orders of magnitude less than cloud GPUs. The extremely limited memory makes deep learning deployment difficult. **Right**: The peak memory and storage usage of widely used deep learning models. ResNet-50 exceeds the resource limit on microcontrollers by $100\times$, MobileNet-V2 exceeds by $20\times$. Even the int8 quantized MobileNetV2 requires $5.3\times$ larger memory and can't fit a microcontroller.

| | Cloud AI (NVIDIA V100) | Mobile AI (iPhone 11) | Tiny AI (STM32F746) | ResNet-50 | MobileNetV2 | MobileNetV2 (int8) |
|---|---|---|---|---|---|---|
| **Memory** | 16 GB → $4\times$ | 4 GB → $3100\times$ | 320 kB | ← gap → 7.2 MB | 6.8 MB | 1.7 MB |
| **Storage** | TB~PB → $1000\times$ | >64 GB → $64000\times$ | 1 MB | ← gap → 102MB | 13.6 MB | 3.4 MB |

library to efficiently manage the tiny resources and fit the tight memory&storage budget. Existing efficient network design [25, 43, 48] and neural architecture search methods [44, 6, 47, 5] focus on GPU or smartphones, where both memory and storage are abundant. Therefore, they only optimize to reduce FLOPs or latency, and the resulting models cannot fit microcontrollers. There is limited literature [16, 31, 42, 29] that studies machine learning on microcontrollers. However, due to the lack of system-algorithm co-design, they either study tiny-scale datasets (*e.g.*, CIFAR or sub-CIFAR level), which are far from real-life use case, or use weak neural networks that cannot achieve decent performance.

In this paper, we propose MCUNet, a system-model co-design framework that enables ImageNet-scale deep learning on off-the-shelf microcontrollers. To handle the scarce on-chip memory on microcontrollers, we jointly optimize the deep learning model design (TinyNAS) and the inference library (TinyEngine) to reduce the memory usage. TinyNAS is a two-stage neural architecture search (NAS) method that can handle the tiny and diverse memory constraints on various microcontrollers. The performance of NAS highly depends on the search space [38], yet there is little literature on the search space design heuristics at the tiny scale. TinyNAS addresses the problem by first optimizing the search space automatically to fit the tiny resource constraints, then performing neural architecture search in the optimized space. Specifically, TinyNAS generates different search spaces by scaling the input resolution and the model width, then collects the computation FLOPs distribution of satisfying networks within the search space to evaluate its priority. TinyNAS relies on the insight that *a search space that can accommodate higher FLOPs under memory constraint can produce better model*. Experiments show that the optimized space leads to better accuracy of the NAS searched model. To handle the extremely tight resource constraints on microcontrollers, we also need a memory-efficient inference library to eliminate the unnecessary memory overhead, so that we can expand the search space to fit larger model capacity with higher accuracy. TinyNAS is co-designed with TinyEngine to lift the ceiling for hosting deep learning models. TinyEngine improves over the existing inference library with code generator-based compilation method to eliminate memory overhead . It also supports model-adaptive memory scheduling: instead of *layer-wise* optimization, TinyEngine optimizes the memory scheduling according to the *overall* network topology to get a better strategy. Finally, it performs *specialized* computation kernel optimization (*e.g.*, loop tiling, loop unrolling, op fusion, *etc*.) for different layers, which further accelerates the inference.

MCUNet dramatically pushes the limit of deep network performance on microcontrollers. TinyEngine reduces the peak memory usage by $3.4\times$ and accelerates the inference by $1.7$-$3.3\times$ compared to TF-Lite and CMSIS-NN, allowing us to run a larger model. With system-algorithm co-design, MCUNet (TinyNAS+TinyEngine) achieves a record ImageNet top-1 accuracy of 70.7% on an off-the-shelf commercial microcontroller. On visual&audio wake words tasks, MCUNet achieves state-of-the-art accuracy and runs $2.4$-$3.4\times$ faster than existing solutions at $3.7$-$4.1\times$ smaller peak SRAM. For interactive applications, our solution achieves 10 FPS with 91% top-1 accuracy on Speech Commands dataset. Our study suggests that the era of tiny machine learning on IoT devices has arrived.

## 2 Background

Microcontrollers have tight memory: for example, only 320kB SRAM and 1MB Flash for a popular ARM Cortex-M7 MCU STM32F746. Therefore, we have to carefully design the inference library and the deep learning models to fit the tight memory constraints. In deep learning scenarios, SRAM constrains the activation size (read&write); Flash constrains the model size (read-only).

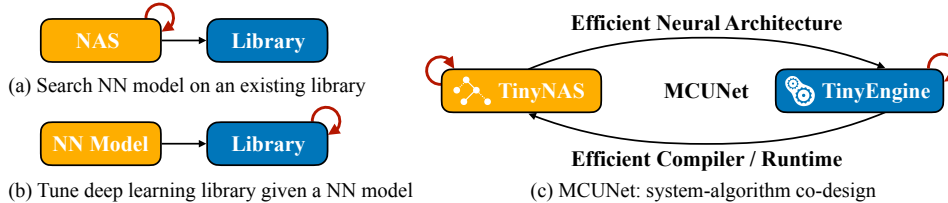

(a) Search NN model on an existing library

(b) Tune deep learning library given a NN model

(c) MCUNet: system-algorithm co-design

**Figure 1.** MCUNet jointly designs the neural architecture and the inference scheduling to fit the tight memory resource on microcontrollers. TinyEngine makes full use of the limited resources on MCU, allowing a larger design space for architecture search. With a larger degree of design freedom, TinyNAS is more likely to find a high accuracy model compared to using existing frameworks.

**Deep Learning Inference on Microcontrollers.** Deep learning inference on microcontrollers is a fast-growing area. Existing frameworks such as TensorFlow Lite Micro [3], CMSIS-NN [28], CMix-NN [8], and MicroTVM [9] have several limitations: 1. Most frameworks rely on an interpreter to interpret the network graph at runtime, which will consume a lot of SRAM and Flash (up to $65\%$ of peak memory) and increase latency by 22%. 2. The optimization is performed at layer-level, which fails to utilize the overall network architecture information to further reduce memory usage.

**Efficient Neural Network Design.** Network efficiency is very important for the overall performance of the deep learning system. One way is to compress off-the-shelf networks by pruning [20, 23, 32, 35, 22, 34] and quantization [19, 50, 39, 49, 13, 11, 45] to remove redundancy and reduce complexity. Tensor decomposition [30, 17, 26] also serves as an effective compression method. Another way is to directly design an efficient and mobile-friendly network [25, 43, 36, 48, 36]. Recently, neural architecture search (NAS) [51, 52, 33, 6, 44, 47] dominates efficient network design.

The performance of NAS highly depends on the quality of the search space [38]. Traditionally, people follow manual design heuristics for NAS search space design. For example, the widely used mobile-setting search space [44, 6, 47] originates from MobileNetV2 [43]: they both use 224 input resolution and a similar base channel number configurations, while searching for kernel sizes, block depths, and expansion ratios. However, there lack standard model designs for microcontrollers with limited memory, so as the search space design. One possible way is to manually tweak the search space for each microcontroller. But manual tuning through trials and errors is labor-intensive, making it prohibitive for a large number of deployment constraints (*e.g.*, STM32F746 has 320kB SRAM/1MB Flash, STM32H743 has 512kB SRAM/2MB Flash, latency requirement 5FPS/10FPS). Therefore, we need a way to automatically optimize the search space for tiny and diverse deployment scenarios.

## 3 MCUNet: System-Algorithm Co-Design

We propose MCUNet, a system-algorithm co-design framework that jointly optimizes the NN architecture (TinyNAS) and the inference scheduling (TinyEngine) in a same loop (Figure 1). Compared to traditional methods that either (a) optimizes the neural network using neural architecture search based on a given deep learning library (*e.g.*, TensorFlow, PyTorch) [44, 6, 47], or (b) tunes the library to maximize the inference speed for a given network [9, 10], MCUNet can better utilize the resources by system-algorithm co-design.

### 3.1 TinyNAS: Two-Stage NAS for Tiny Memory Constraints

TinyNAS is a two-stage neural architecture search method that first optimizes the search space to fit the tiny and diverse resource constraints, and then performs neural architecture search within the optimized space. With an optimized space, it significantly improves the accuracy of the final model.

**Automated search space optimization.** We propose to optimize the search space automatically at *low cost* by analyzing the computation distribution of the satisfying models. To fit the tiny and diverse resource constraints of different microcontrollers, we scale the *input resolution* and the *width multiplier* of the mobile search space [44]. We choose from an input resolution spanning $R = \{48, 64, 80, ..., 192, 208, 224\}$ and a width multiplier $W = \{0.2, 0.3, 0.4, ..., 1.0\}$ to cover a wide spectrum of resource constraints. This leads to $12 \times 9 = 108$ possible search space configurations $S = W \times R$. Each search space configuration contains $3.3 \times 10^{25}$ possible sub-networks. Our goal is to find the best search space configuration $S^*$ that contains the model with the highest accuracy while satisfying the resource constraints.

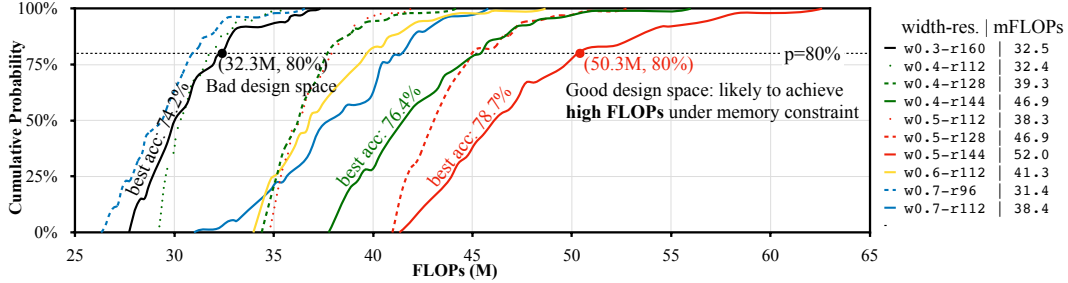

**Figure 2.** TinyNAS selects the best search space by analyzing the FLOPs CDF of different search spaces. Each curve represents a design space. Our insight is that the design space that is more likely to produce *high FLOPs* models under the memory constraint gives higher model capacity, thus more likely to achieve high accuracy. For the solid **red** space, the top 20% of the models have >50.3M FLOPs, while for the solid **black** space, the top 20% of the models only have >32.3M FLOPs. Using the solid **red** space for neural architecture search achieves 78.7% final accuracy, which is 4.5% higher compared to using the **black** space. The legend is in format: `w{width}-r{resolution}|{mean FLOPs}`.

Finding $S^*$ is non-trivial. One way is to perform neural architecture search on each of the search spaces and compare the final results. But the computation would be astronomical. Instead, we evaluate the quality of the search space by randomly sampling $m$ networks from the search space and comparing the distribution of satisfying networks. Instead of collecting the Cumulative Distribution Function (CDF) of each satisfying network's *accuracy* [37], which is computationally heavy due to tremendous training, we only collect the CDF of *FLOPs* (see Figure 2). The intuition is that, within the same model family, the accuracy is usually positively related to the computation [7, 22]. A model with larger computation has a larger capacity, which is more likely to achieve higher accuracy. We further verify the the assumption in Section 4.5.

As an example, we study the best search space for ImageNet-100 (a 100 class classification task taken from the original ImageNet) on STM32F746. We show the FLOPs distribution CDF of the top-10 search space configurations in Figure 2. We sample $m = 1000$ networks from each space and use TinyEngine to optimize the memory scheduling for each model. We only keep the models that satisfy the memory requirement at the best scheduling. To get a quantitative evaluation of each space, we calculate the average FLOPs for each configuration and choose the search space with the largest average FLOPs. For example, according to the experimental results on ImageNet-100, using the solid red space (average FLOPs 52.0M) achieves 2.3% better accuracy compared to using the solid green space (average FLOPs 46.9M), showing the effectiveness of automated search space optimization. We will elaborate more on the ablations in Section 4.5.

**Resource-constrained model specialization.** To specialize network architecture for various microcontrollers, we need to keep a low neural architecture search cost. After search space optimization for each memory constraint, we perform one-shot neural architecture search [4, 18] to efficiently find a good model, reducing the search cost by 200× [6]. We train one super network that contains all the possible sub-networks through *weight sharing* and use it to estimate the performance of each sub-network. We then perform evolution search to find the best model within the search space that meets the on-board resource constraints while achieving the highest accuracy. For each sampled network, we use TinyEngine to optimize the memory scheduling to measure the optimal memory usage. With such kind of co-design, we can efficiently fit the tiny memory budget. The details of super network training and evolution search can be found in the supplementary.

### 3.2 TinyEngine: A Memory-Efficient Inference Library

Researchers used to assume that using different deep learning frameworks (libraries) will only affect the *inference speed* but not the *accuracy*. However, this is not the case for TinyML: the efficiency of the inference library matters a lot to both the latency and accuracy of the searched model. Specifically, a good inference framework will make full use of the limited resources in MCU, avoiding waste of memory, and allow a larger search space for architecture search. With a larger degree of design freedom, TinyNAS is more likely to find a high accuracy model. Thus, TinyNAS is co-designed with a memory-efficient inference library, TinyEngine.

**Code generator-based compilation.** Most existing inference libraries (*e.g.*, TF-Lite Micro, CMSIS-NN) are interpreter-based. Though it is easy to support cross-platform development, it

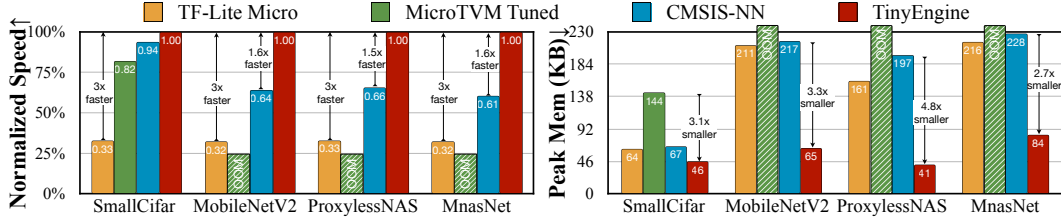

**Figure 3.** TinyEngine achieves higher inference efficiency than existing inference frameworks while reducing the memory usage. **Left**: TinyEngine is 3× and 1.6× faster than TF-Lite Micro and CMSIS-NN, respectively. Note that if the required memory exceeds the memory constraint, it is marked with "OOM" (out of memory). **Right**: By reducing the memory usage, TinyEngine can run various model designs with tiny memory, enlarging the design space for TinyNAS under the limited memory of MCU. Model details in supplementary.

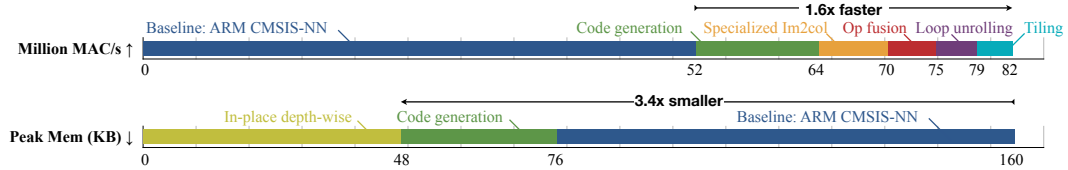

**Figure 4.** TinyEngine outperforms existing libraries by eliminating runtime overheads, specializing each optimization technique, and adopting in-place depth-wise convolution. This effectively enlarges design space for TinyNAS under a given latency/memory constraint.

requires extra memory, the most expensive resource in MCU, to store the meta-information (such as model structure parameters). Instead, TinyEngine only focuses on MCU devices and adopts code generator-based compilation. It not only avoids the time for runtime interpretation, but also frees up the memory usage to allow design and inference of larger models. Compared to CMSIS-NN, TinyEngine reduced memory usage by 2.1× and improve inference efficiency by 22% via code generation, as shown in Figures 3 and 4.

The binary size of TinyEngine is light-weight, making it very memory-efficient for MCUs. Unlike interpreter-based TF-Lite Micro, which prepares the code for *every* operation (e.g., conv, softmax) to support cross-model inference even if they are not used, which has high redundancy. TinyEngine only compiles the operations that are used by a given model into the binary. As shown in Figure 5, such model-adaptive compilation reduces code size by up to 4.5× and 5.0× compared to TF-Lite Micro and CMSIS-NN, respectively.

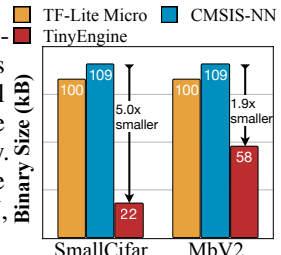

**Figure 5.** Binary size.

**Model-adaptive memory scheduling.** Existing inference libraries schedule the memory for each layer solely based on the layer itself: in the very beginning, a large buffer is designated to store the input activations after im2col; when executing each layer, only one column of the transformed inputs takes up this buffer. This leads to poor input activation reuse. Instead, TinyEngine smartly adapts the memory scheduling to the model-level statistics: the *maximum* memory $M$ required to fit exactly one column of transformed inputs over all the layers $\boldsymbol{L}$,

$$M = \max \left( \text{kernel size}_{L_i}^2 \cdot \text{in channels}_{L_i}; \forall L_i \in \boldsymbol{L} \right). \quad (1)$$

For each layer $L_j$, TinyEngine tries to tile the computation loop nests so that, as many columns can fit in that memory as possible,

$$\text{tiling size of feature map width}_{L_j} = \lfloor M / \left( \text{kernel size}_{L_j}^2 \cdot \text{in channels}_{L_j} \right) \rfloor. \quad (2)$$

Therefore, even for the layers with the same configuration (*e.g.*, kernel size, #in/out channels) in two different models, TinyEngine will provide different strategies. Such adaption fully uses the available memory and increases the input data reuse, reducing the runtime overheads including the memory fragmentation and data movement. As shown in Figure 4, the model-adaptive im2col operation improved inference efficiency by 13%.

**Computation kernel specialization.** TinyEngine *specializes* the kernel optimizations for different layers: loops tiling is based on the kernel size and available memory, which is different for each layer; and the inner loop unrolling is also specialized for different kernel sizes (*e.g.*, 9 repeated code

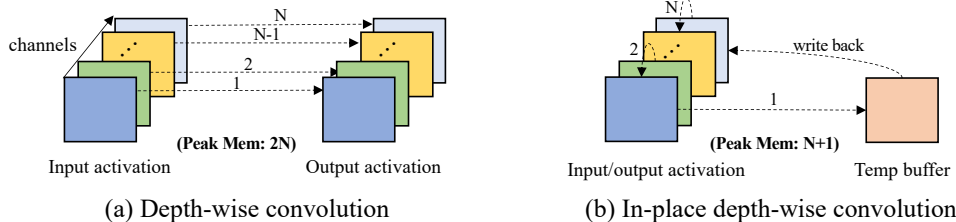

(a) Depth-wise convolution                 (b) In-place depth-wise convolution

**Figure 6.** TinyEngine reduces peak memory by performing in-place depth-wise convolution. **Left:** Conventional depth-wise convolution requires 2N memory footprint for activations. **Right:** in-place depth-wise convolution reduces the memory of depth-wise convolutions to N+1. Specifically, the output activation of the first channel is stored in a temporary buffer. Then, for each following channel, the output activation overwrites the input activation of its previous channel. Finally, the output activation of the first channel stored in the buffer is written back to the input activation of the last channel.

segments for 3×3 kernel, and 25 for 5×5 ) to eliminate the branch instruction overheads. Operation fusion is performed for Conv+Padding+ReLU+BN layers. These specialized optimization on the computation kernel further increased the inference efficiency by 22%, as shown in Figure 4.

https://www.overleaf.com/project/5ebe0deb53b27f000157ca2b

**In-place depth-wise convolution**    We propose *in-place* depth-wise convolution to further reduce peak memory. Different from standard convolutions, depth-wise convolutions do not perform filtering across channels. Therefore, once the computation of a channel is completed, the input activation of the channel can be overwritten and used to store the output activation of another channel, allowing activation of depth-wise convolutions to be updated in-place as shown in Figure 6. This method reduces the measured memory usage by 1.6× as shown in Figure 4.

## 4   Experiments

### 4.1   Setups

**Datasets.**    We used 3 datasets as benchmark: ImageNet [14], Visual Wake Words (VWW) [12], and Speech Commands (V2) [46]. ImageNet is a standard large-scale benchmark for image classification. VWW and Speech Commands represent popular microcontroller use-cases: VWW is a vision based dataset identifying whether a person is present in the image or not; Speech Commands is an audio dataset for keyword spotting (*e.g*., "Hey Siri"), requiring to classify a spoken word from a vocabulary of size 35. Both datasets reflect the always-on characteristic of microcontroller workload. We did not use datasets like CIFAR [27] since it is a small dataset with a limited image resolution ($32 \times 32$), which cannot accurately represent the benchmark model size or accuracy in real-life cases.

During neural architecture search, in order not to touch the validation set, we perform validation on a small subset of the training set (we split 10,000 samples from the training set of ImageNet, and 5,000 from VWW). Speech Commands has a separate validation&test set, so we use the validation set for search and use the test set to report accuracy. The training details are in the supplementary material.

**Model deployment.**    We perform int8 linear quantization to deploy the model. We deploy the models on microcontrollers of diverse hardware resource, including STM32F412 (Cortex-M4, 256kB SRAM/1MB Flash), STM32F746 (Cortex-M7, 320kB SRAM/1MB Flash), STM32F765 (Cortex-M7, 512kB SRAM/1MB Flash), and STM32H743 (Cortex-M7, 512kB SRAM/2MB Flash). By default, we use STM32F746 to report the results unless otherwise specified. All the latency is normalized to STM32F746 with 216MHz CPU.

### 4.2   Large-Scale Image Recognition on Tiny Devices

With our system-algorithm co-design, we achieve record high accuracy (70.7%) on large-scale ImageNet recognition on microcontrollers. We co-optimize TinyNAS and TinyEngine to find the best runnable network. We compare our results to several baselines. We generate the *best scaling* of MobileNetV2 [43] (denoted as **S-MbV2**) and ProxylessNAS Mobile [6] (denoted as **S-Proxyless**) by compound scaling down the width multiplier and the input resolution until they meet the memory requirement. We train and evaluate the performance of all the satisfying scaled-down models on the

**Table 2.** System-algorithm co-design (TinyEngine + TinyNAS) achieves the highest ImageNet accuracy of models runnable on a microcontroller.

| Library \ Model | S-MbV2 | S-Proxyless | TinyNAS |
|---|---|---|---|
| CMSIS-NN [28] | 35.2% | 49.5% | 55.5% |
| TinyEngine | 47.4% | 56.4% | **61.8%** |

**Table 3.** MCUNet outperforms the baselines at various latency requirements. Both TinyEngine and TinyNAS bring significant improvement on ImageNet.

| Latency Constraint | N/A | 5FPS | 10FPS |
|---|---|---|---|
| S-MbV2+CMSIS | 39.7% | 39.7% | 28.7% |
| S-MbV2+TinyEngine | 47.4% | 41.6% | 34.1% |
| MCUNet | **61.8%** | **49.9%** | **40.5%** |

**Table 4.** MCUNet can handle diverse hardware resource on different MCUs. It outperforms [42] without using advanced mixed-bit quantization (8/4/2-bit) policy under different resource constraints, achieving a record ImageNet accuracy (>70%) on microcontrollers.

| | Quantization | STM32F412 (256kB, 1MB) | STM32F746 (320kB, 1MB) | STM32F765 (512kB, 1MB) | STM32H743 (512kB, 2MB) |
|---|---|---|---|---|---|
| **Rusci *et al*.** [42] | Mixed | 60.2% | - | 62.9% | 68.0% |
| **MCUNet** | 4-bit | **62.0%** | **63.5%** | **65.9%** | **70.7%** |

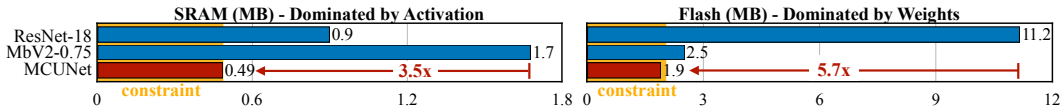

**Figure 7.** MCUNet reduces the the SRAM memory by **3.5×** and Flash usage by **5.7×** compared to MobileNetV2 and ResNet-18 (8-bit), while achieving better accuracy (70.7% *vs*. 69.8% ImageNet top-1).

Pareto front [†], and then report the highest accuracy as the baseline. The former is an efficient manually designed model, the latter is a state-of-the-art NAS model. We did not use MobileNetV3 [24]-alike models because the hard-swish activation is not efficiently supported on microcontrollers.

**Co-design brings better performance.** Both the inference library and the model design help to fit the resource constraints of microcontrollers. As shown in Table 2, when running on a tight budget of 320kB SRAM and 1MB Flash, the optimal scaling of MobileNetV2 and ProxylessNAS models only achieve 35.2% and 49.5% top-1 accuracy on ImageNe using CMSIS-NN [28]. With TinyEngine, we can fit larger models that achieve higher accuracy of 47.4% and 56.4%; with TinyNAS, we can specialize a more accurate model under the tight memory constraints to achieve 55.5% top-1 accuracy. Finally, with system-algorithm co-design, MCUNet further advances the accuracy to 61.8%, showing the advantage of joint optimization.

Co-design improves the performance at various latency constraints (Table 3). TinyEngine accelerates inference to achieve higher accuracy at the same latency constraints. For the optimal scaling of MobileNetV2, TinyEngine improves the accuracy by 1.9% at 5 FPS setting and 5.4% at 10 FPS. With MCUNet co-design, we can further improve the performance by 8.3% and 6.4%.

**Diverse hardware constraints & lower bit precision.** We used int8 linear quantization for both weights and activations, as it is the industrial standard for faster inference and usually has negligible accuracy loss without fine-tuning. We also performed 4-bit linear quantization on ImageNet, which can fit larger number parameters. The results are shown in Table 4. MCUNet can handle diverse hardware resources on different MCUs with Cortex-M4 (F412) and M7 (F746, F765, H743) core. Without mixed-precision, we can already outperform the existing state-of-the-art [42] on microcontrollers, showing the effectiveness of system-algorithm co-design. We believe that we can further advance the Pareto curve in the future with mixed precision quantization.

Notably, our model achieves a record ImageNet top-1 accuracy of 70.7% on STM32H743 MCU. To the best of our knowledge, we are the first to achieve > 70% ImageNet accuracy on off-the-shelf commercial microcontrollers. Compared to ResNet-18 and MobileNetV2-0.75 (both in 8-bit) which achieve a similar ImageNet accuracy (69.8%), our MCUNet reduces the the memory usage by 3.5× and the Flash usage by 5.7× (Figure 7) to fit the tiny memory size on microcontrollers.

---

[†]*e.g*., if we have two models (w0.5, r128) and (w0.5, r144) meeting the constraints, we only train and evaluate (w0.5, r144) since it is strictly better than the other; if we have two models (w0.5, r128) and (w0.4, r144) that fits the requirement, we train both networks and report the higher accuracy.

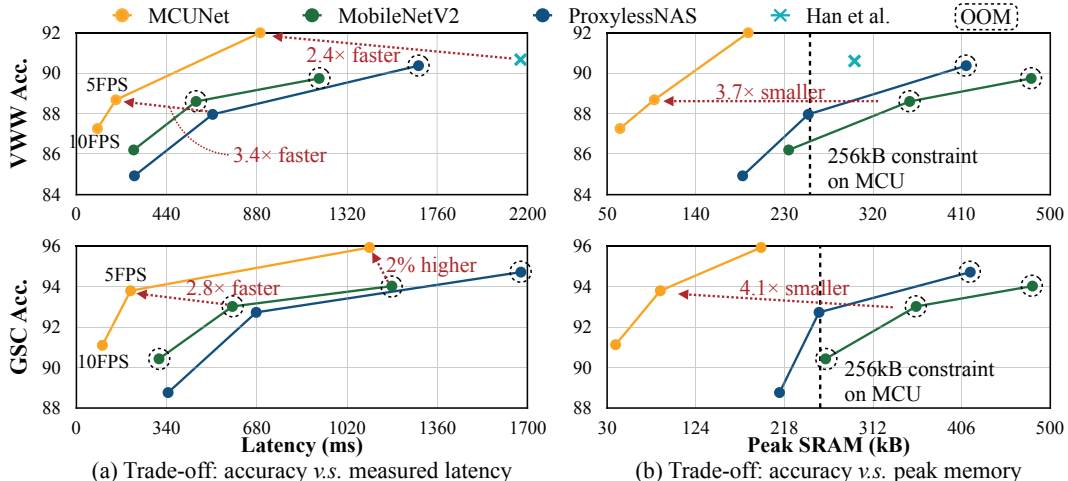

(a) Trade-off: accuracy *v.s.* measured latency        (b) Trade-off: accuracy *v.s.* peak memory

**Figure 8.** Accuracy *vs.* latency/SRAM memory trade-off on VWW (**top**) and Speech Commands (**down**) dataset. MCUNet achieves better accuracy while being 2.4-3.4× faster at 3.7-4.1× smaller peak SRAM.

**Table 5.** MCUNet improves the detection mAP by 20% on Pascal VOC under 512kB SRAM constraint. With MCUNet, we are able to fit a model with much larger capacity and computation FLOPs at a smaller peak memory. MobileNet-v2 + CMSIS-NN is bounded by the memory consumption: it can only fit a model with 34M FLOPs even when the peak memory slightly exceeds the budget, leading to inferior detection performance.

|  | resolution | FLOPs | #Param | peak SRAM | mAP |
|---|---|---|---|---|---|
| MbV2+CMSIS | 128 | 34M | 0.87M | 519kB (OOM) | 31.6% |
| MCUNet | 224 | 168M | 1.20M | 466kB | **51.4%** |

## 4.3 Visual&Audio Wake Words

We benchmarked the performance on two wake words datasets: Visual Wake Words [12] (VWW) and Google Speech Commands (denoted as GSC) to compare the accuracy-latency and accuracy-peak memory trade-off. We compared to the optimally scaled MobileNetV2 and ProxylessNAS running on TF-Lite Micro. The results are shown in Figure 8. MCUNet significantly advances the Pareto curve. On VWW dataset, we can achieve higher accuracy at 2.4-3.4× faster inference speed and 3.7× smaller peak memory. We also compare our results to the previous first-place solution on VWW challenge [1] (denoted as Han *et al.*). We scaled the input resolution to tightly fit the memory constraints of 320kB and re-trained it under the same setting like ours. We find that MCUNet achieves 2.4× faster inference speed compared to the previous state-of-the-art. Interestingly, the model from [1] has a much smaller peak memory usage compared to the biggest MobileNetV2 and ProxylessNAS model, while having a higher computation and latency. It also shows that a smaller peak memory is the key to success on microcontrollers.

On the Speech Commands dataset, MCUNet achieves a higher accuracy at 2.8× faster inference speed and 4.1× smaller peak memory. It achieves 2% higher accuracy compared to the largest MobileNetV2, and 3.3% improvement compared to the largest runnable ProxylessNAS under 256kB SRAM constraint.

## 4.4 Object Detection on MCUs

To show the generalization ability of MCUNet framework across different tasks, we apply MCUNet to object detection. Object detection is particularly challenging for memory-limited MCUs: a high-resolution input is usually required to detect the relatively small objects, which will increase the peak performance significantly. We benchmark the object detection performance of our MCUNet and scaled MobileNetV2+CMSIS-NN on on Pascal VOC [15] dataset. We used YOLOv2 [40] as the detector; other more advanced detectors like YOLOv3 [41] use multi-scale feature maps to generate the final prediction, which has to keep intermediate activations in the SRAM, increasing the peak memory by a large margin. The results on H743 are shown in Table 5. Under tight memory budget (only 512kB SRAM and 2MB Flash), MCUNet significantly improves the mAP by 20%, which makes AIoT applications more accessible.

| R-18@224 | Rand Space | Huge Space | Our Space |
|---|---|---|---|
| **Acc.** | 80.3% | 74.7±1.9% | 77.0% | **78.7%** |

**Table 6.** Our search space achieves the best accuracy, closer to ResNet-18@224 resolution (OOM). Randomly sampled and a huge space (contain many configs) leads to worse accuracy.

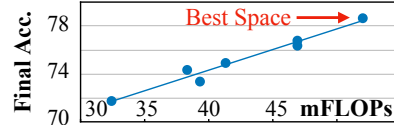

**Figure 9.** Search space with higher mean FLOPs leads to higher final accuracy.

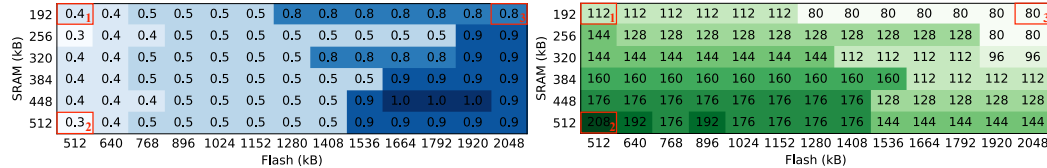

(a) Best width setting.    (b) Best resolution setting.

**Figure 10.** Best search space configurations under different SRAM and Flash constraints.

## 4.5 Analysis

**Search space optimization matters.**    Search space optimization significantly improves the NAS accuracy. We performed an ablation study on ImageNet-100, a subset of ImageNet with 100 randomly sampled categories. The distribution of the top-10 search spaces is shown in Figure 2. We sample several search spaces from the top-10 search spaces and perform the whole neural architecture search process to find the best model inside the space that can fit 320kB SRAM/1MB Flash.

We compare the accuracy of the searched model using different search spaces in Table 6. Using the search space configuration found by our algorithm, we can achieve 78.7% top-1 accuracy, closer to ResNet-18 on 224 resolution input (which runs out of memory). We evaluate several randomly sampled search spaces from the top-10 spaces; they perform significantly worse. Another baseline is to use a very large search space supporting variable resolution (96-176) and variable width multipliers (0.3-0.7). Note that this "huge space" contains the best space. However, it fails to get good performance. We hypothesize that using a super large space increases the difficulty of training super network and evolution search. We plot the relationship between the accuracy of the final searched model and the mean FLOPs of the search space configuration in Figure 9. We can see a clear positive relationship, which backs our algorithm.

**Sensitivity analysis on search space optimization.**    We inspect the results of search space optimization and find some interesting patterns. The results are shown in Figure 10. We vary the SRAM limit from 192kB to 512kB and Flash limit from 512kB to 2MB, and show the chosen width multiplier and resolution. Generally, with a larger SRAM to store a larger activation map, we can use a higher input resolution; with a larger Flash to store a larger model. we can use a larger width multiplier. When we increase the SRAM and keep the Flash from point 1 to point 2 (red rectangles), the width is not increased as Flash is small; the resolution increases as the larger SRAM can host a larger activation. From point 1 to 3, the width increases, and the resolution actually decreases. This is because a larger Flash hosts a wider model, but we need to scale down the resolution to fit the small SRAM. Such kind of patterns is non-trivial and hard to discover manually.

**Evolution search.**    The curve of evolution search on different inference library is in Figure 11. The solid line represents the average value, while the shadow shows the range of (min, max) accuracy. On TinyEngine, evolution clearly outperforms random search, with 1% higher best accuracy. The evolution on CMSIS-NN leads to much worse results due to memory inefficiency: the library can only host a smaller model compared to TinyEngine, which leads to lower accuracy.

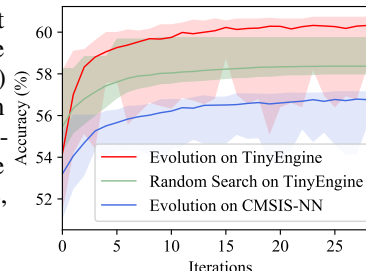

**Figure 11.** Evolution progress.

## 5 Conclusion

We propose MCUNet to jointly design the neural network architecture (TinyNAS) and the inference library (TinyEngine), enabling deep learning on tiny hardware resources. We achieved a record ImageNet accuracy (70.7%) on off-the-shelf microcontrollers, and accelerated the inference of wake word applications by 2.4-3.4×. Our study suggests that the era of always-on tiny machine learning on IoT devices has arrived.

## Statement of Broader Impacts

Our work is expected to enable tiny-scale deep learning on microcontrollers and further democratize deep learning applications. Over the years, people have brought down the cost of deep learning inference from $5,000 workstation GPU to $500 mobile phones. We now bring deep learning to microcontrollers costing $5 or even less, which greatly expands the scope of AI applications, making AI much more accessible.

Thanks to the low cost and large quantity (250B) of commercial microcontrollers, we can bring AI applications to every aspect of our daily life, including personalized healthcare, smart retail, precision agriculture, smart factory, *etc*. People from rural and under-developed areas without Internet or high-end hardware can also enjoy the benefits of AI. Our method also helps combat COVID-19 by providing affordable deep learning solutions detecting face masks and people gathering on edge devices without sacrificing privacy.

With these always-on low-power microcontrollers, we can process raw sensor data right at the source. It helps to protect privacy since data no longer has to be transmitted to the cloud but processed locally.

## Acknowledgments

We thank MIT Satori cluster for providing the computation resource. We thank MIT-IBM Watson AI Lab, Qualcomm, NSF CAREER Award #1943349 and NSF RAPID Award #2027266 for supporting this research.

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
