[Supplementary Material]

# Supplementary Materials for
# MCUNet: Tiny Deep Learning on IoT Devices

## A    Demo Video

We release a demo video of MCUNet running visual wake words. MCUNet achieves **12%** higher accuracy and **2.6×** faster speed compared to MobilenetV1 on TF-Lite Mircro [1]. Please refer to the anonymous link https://www.youtube.com/watch?v=sDRxruEbpRM for details. We also provide a low-resolution version in the attachment: mcunet_demo_360p.mp4.

Note that we show the actual frame rate in the demo video, which includes frame capture overhead from the camera (around 30ms per frame). It slows down the inference from 10 FPS to 7.4 FPS for our model.

## B    Model Release

We release MCUNet models in TF-Lite format in the supplementary materials. Please refer to mcunet-eval/README.md for details. We will also release the TinyEngine and TinyNAS codebase once cleaned up.

## C    Missing Connected Curves in Figure 6

The connected curves in Figure 6(a) of the paper is missing due to LaTeX compilation error. The fixed version is below:

**Figure S1.** Accuracy *vs*. latency/SRAM memory trade-off on VWW (**top**) and Speech Commands (**down**) dataset. MCUNet achieves better accuracy while being 2.4-3.4× faster at 2.2-2.6× smaller peak SRAM.

## D  Binary size of Inference Libraries

The binary size of TinyEngine is light-weight, making it very memory-efficient for MCUs. Unlike interpreter-based TF-Lite Micro, which prepares *every* operation (e.g., conv, softmax) to support cross-model inference even if they are not used, which has high redundancy. TinyEngine only compiles the operations that are used by a given model into the binary. As shown in Figure S2, such model-adaptive compilation reduces code size by up to $4.5\times$ and $5.0\times$ compared to TF-Lite Micro and CMSIS-NN, respectively.

**Figure S2.** Binary size of different inference libraries. TinyEngine requires less binary size compared to TFLite-Micro and CMSIS-NN.

## E  Design Cost

There are billions of IoT devices with drastically different constraints, which requires different search spaces and model specialization. Therefore, keeping a low design cost is important.

MCUNet is efficient in terms of neural architecture design cost. The search space optimization process takes negligible cost since no training or testing is required (it takes around 2 CPU hours to collect all the FLOPs statistics). The process needs to be done only once and can be reused for different constraints (*e.g.*, covered two MCU devices and 4 memory constraints in Table 4). The one-shot neural architecture search is far more efficient compared to traditional neural architecture search method: it takes 40,000 GPU hours for MnasNet [8] to design a model, while MCUNet only takes 300 GPU hours, reducing the search cost by $133\times$. With MCUNet, we reduce the $CO_2$ emission from 11.4 lbs to 0.08 lbs per model (Figure S3).

**Figure S3.** Total $CO_2$ emission (lbs) for model design. MCUNet saves the design cost by orders of magnitude, allowing model specialization for different deployment scenarios.

## F  Resource-Constrained Model Specialization Details

For all the experiments in our paper, we used the same training recipe for neural architecture search to keep a fair comparison.

**Super network training.**  We first train a super network to contain all the sub-networks in the search space through *weight sharing*. Our search space is based on the widely-used mobile search space [8, 3, 10, 2] and supports variable kernel sizes for depth-wise convolution (3/5/7), variable expansion ratios for inverted bottleneck (3/4/6) and variable stage depths (2/3/4). The input resolution and width multiplier is chosen from search the space optimization technique proposed in section 3.1. The number of possible sub-networks that TinyNAS can cover in the search space is large: $2 \times 10^{19}$.

To speed up the convergence, we first train the largest sub-network inside the search space (all kernel size 7, all expansion ratio 6, all stage depth 4). We then use the trained weights to initialize the super network. Following [2], we sort the channels weights according to their importance (we used L-1 norm to measure the importance [6]), so that the most important channels are ranked higher. Then

we train the super network to support different sub-networks. For each batch of data, we randomly sample 4 sub-networks, calculate the loss, backpropagate the gradients for each sub-network, and update the corresponding weights. For weight sharing, when select a smaller kernel, *e.g.*, kernel size 3, we index the central $3 \times 3$ window from the $7 \times 7$ kernel; when selecting a smaller expansion ratio, *e.g.* 3, we index the first $3n$ channels from the $6n$ channels ($n$ is #block input channels), as the weights are already sorted according to importance; when using a smaller stage depth, *e.g.* 2, we calculate the first 2 blocks inside the stage the skip the rest. Since we use a fixed order when sampling sub-networks, we keep the same sampling manner when evaluating their performance.

**Evolution search.** After super-network training, we use evolution to find the best sub-network architecture. We use a population size of 100. To get the first generation of population, we randomly sample sub-networks and keep 100 satisfying networks that fit the resource constraints. We measure the accuracy of each candidate on the independent validation set split from the training set. Then, for each iteration, we keep the top-20 candidates in the population with highest accuracy. We use crossover to generate 50 new candidates, and use mutation with probability 0.1 to generate another 50 new candidates, which form a new generation of size 100. We measure the accuracy of each candidate in the new generation. The process is repeated for 30 iterations, and we choose the sub-network with the highest validation accuracy.

# G    Training&Testing Details

**Training.** The super network is trained on the training set excluding the split validation set. We trained the network using the standard SGD optimizer with momentum 0.9 and weight decay 5e-5. For super network training, we used cosine annealing learning rate [7] with a starting learning rate 0.05 for every 256 samples. The largest sub-network is trained for 150 epochs on ImageNet [5], 100 epochs on Speech Commands [9] and 30 epochs on Visual Wake Words [4] due to different dataset sizes. Then we train the super network for twice training epochs by randomly sampling sub-networks.

**Validation.** We evaluate the performance of each sub-network on the independent validation set split from the training set in order not to over-fit the real validation set. To evaluate each sub-network's performance during evolution search, we index and inherit the partial weights from the super network. We re-calibrate the batch normalization statistics (moving mean and variance) using 20 batches of data with a batch size 64. To evaluate the final performance on the real validation set, we also fine-tuned the best sub-network for 100 epochs on ImageNet.

**Quantization.** For most of the experiments (except Table 4), we used TensorFlow's int8 quantization (both activation and weights are quantized to int8). We used post-training quantization without fine-tuning which can already achieve negligible accuracy loss. We also reported the results of 4-bit integer quantization (weight and activation) on ImageNet (Table 4 of the paper). In this case, we used quantization-aware fine-tuning for 25 epochs to recover the accuracy.