[Reviews · NeurIPS 2020]

Review 1

Summary and Contributions: This paper proposes a method that co-designs TinyNAS with TinyEngine, which deploys the deep neural networks on the commercial microcontroller STM32 and achieves promising results on ImageNet. The proposed MCUNet is the first one to achieve >70% ImageNet accuracy on the off-the-shelf commercial microcontroller. This is a very significative direction for research of efficient deep learning and AutoML.

Strengths: 1.Good performance. 2. The method is easy to follow and makes sense. 3. This paper proposes a two-stage NAS, TinyNAS, to first optimize the search space and then perform NAS to meet the constraints, which searches for architectures under a wide space with low cost. 4. The proposed TinyEngine introduces model-adaptive memory scheduling and computation kernel specialization which greatly improves inference efficiency and enlarges the design space in turn. TinyEngine is used for co-design with TinyNAS. As a consequence, both the architecture and accelerator are optimized. Moreover, the memory scheduling is adapted according to the overall network, not only the layer-wise optimization, which is benefit for the memory usage reduction. 5. The final deployed model achieves great performance on ImageNet and Visual&Audio Wake Words. Especially, MCUNet achieves 70.2% top-1 accuracy on ImageNet on the STM32 MCU. The result is very promising as the capacity of STM32 is very small.

Weaknesses: 1. The co-designing mechanism is the core contribution of the paper but the detailed process is not described clearly. It is suggested to provide an elaborate diagram or pseudocode to introduce the whole framework. 2. MCUNet achieves convincing results on STM32. It is meaningful to explore its generalization ability. Please demonstrate the potential/ability of MCUNet being deployed on more scenarios and tasks, e.g., more devices or tasks like object detection, semantic segmentation. 3.The method is described as a co-design scheme. But the experiments failed to highlight the improvements of the co-design scheme, when compared to those single design schemes. 4.The author introduce the tiny engine with the motivation that "TinyEngine optimizes the memory scheduling according to the overall network topology to get a better strategy." But it is unclear whether the overall network topology indeed is the main reason for the huge improvements. The authors need to add supporting results or supporting articles. 5. Some typo issues need to be checked, e.g., "to achieves" in line 14.

Correctness: The claims and method are correct. The experiments are performed comprehensively and the empirical methodology are correct.

Clarity: The co-designing details are not so clear. The other parts of the paper are described clearly and are easy to follow.

Relation to Prior Work: The relation to prior work is discussed clearly. Though many previous works attempt to optimize neural architectures on customized hardware platform. Rare of them co-design the architecture with the accelerator. This brings more probabilities for TinyML being applied in wide fields.

Reproducibility: Yes

Additional Feedback: After reading the rebuttal and comments from other reviewers, I would like to keep my original rating. Regarding some details that need to be fully illustrated in this paper. The proposed method is meaningful that may have a great impact on the TinyAI .


Review 2

Summary and Contributions: This paper proposes a joint design of neural architecture search and inference engine for MCUs. The NAS part scales the input resolution and width multiplier to fit the model to the limited resource. A novel CDF analysis has been proposed to prune the search space and reduce the overhead of NAS. The inference engine adapts the maximum memory based on the whole network and leverages the adaptive resource for layer-wise execution.

Strengths: This paper proposes a novel CDF analysis to resolve the additional overhead of searching input resolution for MCUs. The empirical evaluations have illustrated the superior performance of the design. Authors have conducted extensive experiments to validate the performance gain of their NAS algorithm as well as the inference engine. The video in the supplementary material is a good demo for the paper.

Weaknesses: I assume that the microTVM in Figure 3 uses the auto-tuning to optimize the performance. Could authors provide more discussion about why the auto-tuning in TVM fails to works on MCUs (since TinyEngine clearly outperforms it in Figure 3)

Correctness: The claims and method looks correct. The video in the supplementary material illsurates the real deployment of MCUNet on STM32F746.

Clarity: The paper is well written.

Relation to Prior Work: This paper has address the contirbutions compared to the previous hardware-aware NAS with evaluaitons.

Reproducibility: Yes

Additional Feedback:


Review 3

Summary and Contributions: To run deep learning on MCU effectively, the paper proposes MCUNet which contains a neural architecture search method (TinyNAS) and a lightweight inference library (TinyEngine). TinyNAS optimizes the search space automatically to fit the tiny resource constraints and then performing the one-shot neural architecture search in the optimized space. TinyEngine uses a code generator-based compilation method to eliminate memory overhead in MCU. It is exciting to read this paper. By TinyNAS & TinyEngine co-design, MCUNet obtains SOTA results on ImageNet classification on MCU, and visual&audio wake words tasks.

Strengths: 1. The paper is of great significance - reaching exciting results on a MCU promotes the development of AIoT. 2. The neural architecture and inference engine co-design idea is novel. By deep experimental analysis, the proposed tricks are proven to be effective. 3. The empirical results are excellent - large improvements are obtained over TensorFlowLite micro and CMSIS-NN. 4. Extensive evaluations on different tasks. It has proven to be effective for large-scale visual recognition and practical applications, and visual&audio wake words tasks.

Weaknesses: A minor issue: How does the TinyEngine scheduling affect TinyNAS is not clear enough. I suggest adding a flowchart to detail the co-design process.

Correctness: Yes, they are correct.

Clarity: Yes, well-enough.

Relation to Prior Work: Yes, prior works are clearly discussed.

Reproducibility: No

Additional Feedback:


Review 4

Summary and Contributions: The authors tackle the problem of tiny ML, with the objective of efficiently deploying deep learning techniques on micro-controllers. The main contributions are: - TinyNAS, a two stage neural architecture search algorithm that first automatically designs a search space adapted to the target platform and then apply NAS techniques to select the best architecture in this space. - TinyEngine, an inference library increasing the speed and reducing the memory required to evaluate a model on a microcontroller unit.

Strengths: The TinyEngine results are impressive, clearly showing its memory and speed improvements over existing libraries using the same model on the same platform. It results in the ability to fit larger models on the device and therefore improves the overall performance (both latency and accuracy) for application in highly constrained environments. The method used to define the search space of tinyNAS is simple, effective and allows to search for architectures that will be very efficient on the specific microcontroller unit targeted. It is, to my knowledge, the first approach that automatically defines the search space to make the most of the resources that will be available at inference time.

Weaknesses: The small amount of space dedicated to the description of the actual NAS procedure is concerning to me. As it is one of the main contributions, I think it should be presented in more detail in the main paper, with a thorough comparison to the other budget-aware NAS methods allowing to assess its performance and make sure that the proposed comparisons are fair. Another concern is the lack of details about the experimental protocol (e.g. detailed description of the datasets, preprocessing used, training times of the different approaches, $m$ used for each experiment) that would make the results difficult to replicate.

Correctness: Is TinyNAS a new neural architecture search algorithm or a smart way of defining efficient search spaces? If it is an actual NAS algorithm, explaining it and running more experiments to compare the effectiveness of the proposed search algorithm (*using the same search spaces*) to other recent methods would be necessary to validate the importance of this contribution. If it is an algorithm allowing to define device-specific search spaces, I think it should be presented as such. Clearly stating that it is not a NAS algorithm and putting the emphasis on the TinyEngine.

Clarity: The paper is overall well written and pleasing to read.

Relation to Prior Work: While inference libraries are not my area of expertise, I think that the TinyEngine is compared to the relevant high-performing inference libraries designed to be used on a microcontroller. The difference between these libraries being explained in section 3.2 and empirically validated in section 4. The relation of the TinyNas algorithm to prior work in neural architecture search is less clear. In the introduction (l.40), it is stated that other approaches focus on applications where memory and storage are abundant and only optimize for FLOPs or latency. However, there exist some NAS approaches that propose to directly optimize the memory footprint of the learned architecture [1, 2]. I think the paper would benefit from a clear comparison of the actual search algorithm (not only the search space design) and other recent NAS methods. [1] Gordon et al. "MorphNet: Fast & Simple Resource-Constrained Structure Learning of DeepNetworks." CVPR. 2018 [2] Veniat et al. "Learning time/memory-efficient deep architectures with budgeted super networks." CVPR. 2018

Reproducibility: No

Additional Feedback: I think that the authors should put more emphasis on the TinyEngine which is an important contribution by itself. Putting the TinyNAS part as an independent work (or secondary contribution) allowed by the improvements provided by the TinyEngine. Rebuttal: The author's feedback addresses and resolves some of the weaknesses I pointed out in my initial review. Given the new results and the additional explanations about their model, I update my evaluation to better recognize the novelty of this work and its potential impact.

[Author Response · NeurIPS 2020]

We thank all the reviewers for their constructive comments. Below are detailed responses.

**R1&R3: Co-design process elaboration.** We provide a simple pseudo-
code in Alg. A due to space limit. We will provide details in the final draft.

**Algorithm A.** The co-design process.

```
# TinyNAS: sample a DNN arch
for arch in arch_space:
  # TinyEngine: find a good schedule
  for schedule in schedule_space:
    # check if satisfy mem. constraints
    if can_fit_memory(arch, schedule):
      # eval acc. and update best arch
      acc = get_valid_acc(arch)
      best_acc = max(best_acc, acc)
      break
```

**R1: More deployment devices and tasks.** MCUNet generalizes well
across different MCU devices with different capacities: we show the Ima-
geNet top-1 accuracy on F746 (320kB SRAM, 1MB Flash) and H743 (512kB
SRAM, 2MB Flash) in Table A, MCUNet consistently outperforms the base-
line by a large margin (up to 20.4%). MCUNet also generalizes beyond
classification to detection. On PASCAL VOC with YOLO, MCUNet signif-
icantly improves the mAP from 31.6% to 51.4% on H743. To the best of our
knowledge, this is the first large-scale object detection experiment on tiny MCU devices.

|  | ImgNet(F746) | ImgNet(H743) | VOC(H743) |
|---|---|---|---|
| MbV2+CMSIS | 39.7% | 53.8% | 31.6% |
| MCUNet | **60.1%** | **65.1%** | **51.4%** |

**Table A.** MCUNet shows consistent improvement across different devices (F746, H743) and tasks (classification, detection).

| | ImageNet Top1 | | |
|---|---|---|---|
| Baseline (MbV2+CMSIS) | 39.7 | | |
| Single (MbV2+TinyEngine) | 43.80 | | |
| Single (TinyNAS+CMSIS) | 55.5 | | |
| Co-design (TinyNAS+TinyEngine) | 60.1 | | |

**Figure A.** MCUNet's co-design scheme outperforms single-design ones on ImageNet classification.

**R1: Improvements from co-design over single-design.** We showed the advantage of the co-design scheme in Table
2 of the original paper, where co-design achieves 4.6% higher accuracy compared to the best single-design result. We
highlight the advantage of the co-design scheme in Figure A. We will make it more clear in the final draft.

**R1: Whether the overall network topology brings major improvement.** Yes, considering the overall network
topology enables specialized im2col, specialized loop tiling and unrolling strategies, which accounts for 49% of the
overall performance boost achieved by TinyEngine.

**R2: Why the auto-tuning in TVM fails to work on MCUs.** MicroTVM's auto-tuning is based on a pre-defined
implementation template. However, the template does not include our advanced optimizations, *e.g.*, scheduling memory
according to the overall network topology. Therefore, auto-tuning cannot match our speedup and memory reduction.

**R4: Contributions of TinyNAS.** We would like to clarify that TinyNAS is novel for the "actual NAS procedure".
TinyML on MCU is a very new area; existing NAS methods *cannot* fit the tight memory constraints. TinyNAS is the
*first* NAS algorithm to enable large-scale deep learning on MCU devices. Since there is no carefully-tweaked design
space like those for mobile phones, we have to start from a huge search space so that it is likely to contain a good
model for various MCUs. The space needs to cover not only the micro-level architecture designs (*e.g.*, kernel size,
expansion ratio) but also the macro-level designs like input resolution and channel widths (Section 3.1). Existing NAS
methods fail to achieve good performance on the huge space (Table 5 in original paper), since the large space makes
weight-sharing difficult and leads to *low* sample efficiency due to the *sparse* search reward. Our TinyNAS overcomes
the search inefficiency with a two-stage search algorithm. The first stage is to shrink/prune the huge search space to a
smaller sub-space, so the reward is no longer sparse, and the sample efficiency is improved. The second stage is to
perform micro-level optimization in the pruned sub-space. Both stages are the "actual NAS procedure"; they work
jointly with TinyEngine to achieve a decent performance, and should not be considered separately.

**R4: Comparison to budget-aware NAS methods.** TinyNAS argues that a two-stage
algorithm that gradually narrows down the search space is important to avoid the sparse
search award. Therefore, a fair comparison needs to start from the same full space. We
modify existing NAS methods to use the same search space under the same memory
constraint as ours. Compared to Single Path One-Shot NAS (SPOS) [17] and Once-For-
All (OFA) [5] on ImageNet-100 (ImgN$_{100}$), TinyNAS outperforms both SOTA methods
(Table B), which verifies the advantage of our two-stage search mechanism. Other NAS
methods (*e.g.*, [6, 44]) cannot handle the macro-level architecture like backbone channel
widths like ours. Therefore, we scale their channels&resolutions to fit the same memory

**Table B.** Compare NAS.

| Method | ImgN$_{100}$ | ImgN$_{1k}$ |
|---|---|---|
| MnasNet [14] | - | 51.8% |
| FBNet [44] | - | 50.6% |
| Proxyless [6] | - | 54.4% |
| SPOS [17] | 75.6% | 53.6% |
| OFA [5] | 77.0% | 54.0% |
| TinyNAS | **78.7%** | **60.1%** |

budget of STM32F746 (320kB). Under the same MobileNet-v2 search space, TinyNAS shows significant advantage on
ImageNet (ImgN$_{1k}$) with up to **9.5%** better top-1 accuracy, which verifies memory-awareness is important for TinyML.

**R4: Existing NAS methods that optimize memory footprint.** The two papers provided by the reviewer do not
optimize the working memory footprint. MorphNet [Gordon *et al.*, 2018] only considers FLOPs and model size as
constraints. Though [Veniat *et al.*, 2018] mentions "memory consumption cost", it actually refers to model size but not
activation memory, which is the bottleneck. Neither explored memory-bounded NAS at tiny MCU scale (<1MB).

**R4: Details about the experimental protocol.** Many experimental protocol details are provided in Section 4.1 and
Section G of the supplementary (*e.g.*, datasets, momentum, weight decay, training epochs). We will add more details to
the main paper in the final version to help reproduction.

**R4: Limited space for NAS.** Both stages are the actual NAS procedure to search a good model from a huge search
space. Therefore, we have dedicated a considerable amount of space for the NAS procedure. Due to the space limit, we
put some of the details of the second stage in the supplementary. We will add it to the main paper in the final version.

[Meta-Review · NeurIPS 2020]

All four knowledgeable referees support acceptance for the contributions, notably co-designing TinyNAS and TinyEngine for deep learning on IoT devices and promising experimental results on ImageNet, and I also recommend acceptance. Please make it sure to appropriately reflect what has been promised through rebuttal such as elaboration on co-design.